# Dietary Intake of Athletes at the World Masters Athletics Championships as Assessed by Single 24 h Recall

**DOI:** 10.3390/nu16040564

**Published:** 2024-02-19

**Authors:** Taylor P. M. Leonhardt, Ainsley Bristol, Natalie McLaurin, Scott C. Forbes, Hirofumi Tanaka, Petra Frings-Meuthen, Dominik Pesta, Jörn Rittweger, Philip D. Chilibeck

**Affiliations:** 1College of Kinesiology, University of Saskatchewan, Saskatoon, SK S7N 5B2, Canada; tpl314@mail.usask.ca (T.P.M.L.); aib690@mail.usask.ca (A.B.); 2Department of Kinesiology and Health Education, The University of Texas at Austin, Austin, TX 78712, USA; nmclaurin@utexas.edu (N.M.); htanaka@austin.utexas.edu (H.T.); 3Faculty of Education, Department of Physical Education Studies, Brandon University, Brandon, MB R7A 6A9, Canada; forbess@brandonu.ca; 4Institute of Aerospace Medicine, German Aerospace Center (DLR), D-51147 Cologne, Germany; petra.frings-meuthen@dlr.de (P.F.-M.); dominik.pesta@dlr.de (D.P.); joern.rittweger@dlr.de (J.R.)

**Keywords:** track and field, aging, elderly, senior, power, endurance, nutrition

## Abstract

Proper dietary intake is important for masters athletes because of the physiological changes that occur with aging and the unique nutritional needs when competing at high levels. We evaluated the dietary intake of masters athletes competing at the World Masters Athletics Championships (outdoor games, Tampere, Finland, 2022, and indoor games, Torun, Poland, 2023). A total of 43 athletes (16 females and 27 males, mean age 59.2 ± 10.3 y, height 168 ± 8 cm, and body mass 62.3 ± 10.8 kg) participating in endurance (n = 21), sprint (n = 16), jumping (2), multi-component (e.g., decathlon; n = 3), and throwing (n = 1) events provided 24 h dietary recalls while participating in the games. Carbohydrate intake was below the recommended levels for endurance athletes. Protein intake was below the recommended levels for masters athletes, except for female athletes involved in power events (i.e., sprinters and jumpers). Other nutrient intakes that were below the recommended levels included vitamins D and E, calcium, potassium, vitamin A (except for female endurance athletes), folate (except for female power athletes), vitamin C for female endurance athletes, vitamin K and fiber for males, and zinc for endurance athletes. We conclude that while competing at world championships, many athletes are not consuming the recommended levels of carbohydrates, protein, and micronutrients. Athletes attending these games would benefit from increased nutritional support.

## 1. Introduction

Proper nutritional intake is important for performance and health in athletes [1]. This is especially true in older (i.e., masters) athletes who have some special requirements due to the physiological changes that come with aging [2]. This may include, for example, a greater requirement compared to younger athletes for protein due to “anabolic resistance” (i.e., reduced muscle protein synthesis in response to training and amino acid consumption), antioxidant micronutrients to protect against cellular damage, and vitamin D and calcium to maintain bone health [2,3].

A recent systematic review of the dietary intake of masters athletes indicated that while dietary intakes were generally higher than the general population, intakes of protein and carbohydrate may be insufficient to meet the demands of athletic performance and the intake of calcium might be lower than recommended for the maintenance of bone health [4]. Athletics (i.e., track and field) places unique demands on masters athletes, ranging from high requirements for muscular power (i.e., sprinters, jumpers, and throwers) to aerobic endurance (i.e., medium and long-distance runners, and racewalkers), as well as multi-discipline events (i.e., decathlon, heptathlon, and pentathlon). Very few studies have evaluated the dietary intake of masters athletes involved in athletics competitions. No studies have determined intakes of specific nutrients, but two have indicated relatively low intake of important foods, such as fruits, vegetables, whole grains, dairy, and fish in male athletes [5,6]. No studies have analyzed the specific macro- and micronutrient intakes of male and female masters athletes involved in athletics, and none have evaluated intakes during competition, when nutritional intake may prove challenging due to travel and competing in a foreign environment.

The purpose of our study was to evaluate the dietary intake of highly competitive masters athletes from both sexes across a variety of endurance and power-based events during world athletics championship competitions. Based on previous research on masters athletes involved in other sports [4] and the challenges of obtaining proper nutrition while traveling, it was hypothesized that masters athletes at world championship athletics competitions would consume less than the recommended levels for many nutrients.

## 2. Materials and Methods

A total of 43 athletes (16 females and 27 males, mean age 59.2 ± 10.3 y, height 168 ± 8 cm, and body mass 62.3 ± 10.8 kg) participating in endurance (n = 21), sprint (n = 16), jumping (n = 2), multi-component (i.e., decathlon, pentathlon, and heptathlon; n = 3), and throwing (n = 1) events provided 24-hour dietary recalls while participating in either the outdoor (Tampere, Finland, 2022) or the indoor (Torun, Poland, 2023) World Masters Athletics Championships. Endurance athletes were classified as those involved in running events ≥800 m or racewalking, and power athletes were classified as those competing in sprinting, jumping, throwing, decathlon, heptathlon, or pentathlon. The classification of events ≥800 m as “endurance” is based on the finding that this is the race distance at which the aerobic energy system begins to predominate, with aerobic energy system contribution estimated at 60% for men and 70% for women [7]. On the other hand, the 400 m was classified as a “sprint” as the anaerobic energy system predominates, with anaerobic energy system contribution estimated at 59% for men and 55% for women [7]. Multi-event athletes (i.e., decathlon, heptathlon, and pentathlon) were classified as power athletes, as only one event in each of these disciplines requires aerobic endurance. The 24 h recalls were analyzed using Food Processor Software (ESHA Research, Version 11.1, Salem, Orlando, FL, USA), which produced an estimate of daily energy, macronutrient, and micronutrient intake. This study was approved by the University of Saskatchewan Biomedical Research Ethics Board (Bio #1834), and all participants signed an informed consent form prior to participating.

Descriptive statistics were calculated as means and standard deviations (SD) for energy intake, macronutrients, and micronutrients for male and female endurance and power athletes. Descriptive results were compared with current recommended values [1,8]. A two-way analysis of variance was used to compare athletic groups (endurance versus power) and sexes (females versus males) and to determine group—sex interactions for each nutrient. Chi-squared tests were used to compare the proportion of athletes who met the recommended intakes from each athletic group (i.e., female endurance, male endurance, female power, and male power athletes). Statistical analyses were run with Statistica 7.0 (Statsoft, Chicago, IL, USA). A *p*-value less than or equal to 0.05 was accepted as the level of significance. 

## 3. Results

Absolute levels of energy, macronutrients, and micronutrients are presented in Table 1, as well as the proportion (%) of athletes meeting the recommended intakes for male and female endurance and power athletes separately and all athletes combined. Figure 1 shows intakes relative to 100% of the recommendation for macronutrients and micronutrients. For macronutrients, carbohydrate intake was lower than recommended by sport nutrition guidelines for endurance athletes [1]. Most groups, except female power athletes, were below the protein intake recommendations for masters athletes [2,3]. Fat and saturated fat intake were at acceptable levels, but omega-3 fatty acid intake was below the level recommended for masters athletes [2,3]. All groups had intakes of vitamins D and E, calcium, and potassium below the recommended levels. All groups except female endurance athletes had an intake of vitamin A below the recommended levels. Other intakes that were below recommended levels included vitamin C for female endurance athletes, zinc for male and female endurance athletes, vitamin K and fiber for male endurance and power athletes, and folate for all except female power athletes (Table 1, Figure 1). Although mean intakes were at recommended levels, the proportion (i.e., percentage) of endurance athletes consuming the recommended levels of vitamins B1 and B6 and iron were relatively low (Table 1).

Power athletes had greater intakes of energy, absolute level of protein (g), folate, potassium, vitamin B1, and vitamin D compared to endurance athletes (*p* < 0.05; Table 1). The only sex difference was that females had a greater intake of carbohydrates relative to body mass compared to males (*p* < 0.05). The proportion (%) of athletes meeting the recommended intakes was different between groups for all nutrients except vitamin B2 and sodium (*p* < 0.05; Table 1). In most cases, female power athletes had the greatest proportion of athletes meeting the recommended intakes for nutrients, except the male endurance athlete group which had the greatest proportion who met the recommended intake for fat (% of kcal), the female endurance athlete group which had the greatest proportion who met recommended intakes for vitamin A and vitamin K, and the male power athlete group which had the greatest proportion who met the recommended intakes for vitamin E, calcium, and iron (Table 1).

## 4. Discussion

The main findings from this study were that masters athletes involved in world athletics championship competitions on average do not consume protein, carbohydrates, and a number of micronutrients at sufficient levels to support optimal performance, as assessed by single 24 h dietary recalls. This indicates that more support might be needed for the athletes when traveling to a foreign location for competitions. This study is unique in that it is the first study to analyze intakes of specific macro- and micronutrients in male and female masters athletes involved in world championship athletic competitions.

Regarding macronutrients, carbohydrate and protein intake may have been at insufficient levels to support optimal performance. Carbohydrates are especially important for endurance athletes for maintenance of muscle glycogen levels. This is important for masters athletes, as older adults may have decreased muscle glycogen [9]. Carbohydrate intake averaged around 3.2 g/kg/d in the endurance sport athletes evaluated in our study. Sport nutrition guidelines recommend intakes of 5–7 g/kg/d of carbohydrates for athletes training at moderate intensity for one hour per day and intakes of 6–10 g/kg/d for athletes training 1–3 h per day at moderate to high intensities [1]. The only other studies of masters track and field athletes evaluating dietary intake were at the European Athletics Championships, which determined low intakes of carbohydrate-containing foods such as fruits and whole-grain bread and cereals [5], and one study at the indoor World Athletics Championships which also measured low consumption of whole-grain foods [6]; both of these studies only included male participants. One previous study in recreational masters marathoners (again including only males) determined that carbohydrate intake was below guidelines during the week of competition and that carbohydrate intake was predictive of performance [10]. A recent meta-analysis of masters athletes involved in other sports determined carbohydrate intakes of 3.5–4.1 g/kg/d for females and 3–5.3 g/kg/d in males [4]. The females from our study had similar intakes (~4.0 g/kg), whereas the intake for males in our study was on the lower end of this range (~3.0 g/kg). These intakes are still below guidelines [1], indicating that the athletes in our study may have found it challenging to consume sufficient carbohydrates or sufficient calories during competition when traveling to a foreign environment. We can only speculate on why carbohydrate intake during competition was relatively low. This may be due to limited high-carbohydrate food available at the stadium or perhaps that the athletes had insufficient time to consume high-carbohydrate foods while competing.

Fiber intake is important to lower blood lipids and improve the gut microbiota in older adults [11]. Females, but not males, in our study achieved the recommended intake levels of dietary fiber. Sport nutrition guidelines recommend lower fiber intake for endurance athletes close to competition to avoid stomach discomfort [1] and this might have influenced the intake in our athletes who were competing at the time of dietary analyses.

Protein intake of our athletes was lower than recommended for masters athletes, except for the female power athletes (Table 1). Anabolic resistance may increase with age, as older adults have lower muscle protein synthesis in response to muscle contraction [12,13] and amino acid intake [14] and therefore require a greater amount of protein to stimulate muscle protein synthesis [14]. Most recommendations for masters athletes involve protein intakes of 1.5–1.6 g/kg/d [2,3,15], especially when energy intake is low (as with the athletes in our study), in order to preserve lean tissue mass [2,3]. An observational study of masters athletes indicated that those with higher protein intake (i.e., 1.34 vs. 1.21 g/kg/d) had greater muscle strength [16], implying that higher intake is beneficial for power athletes. Higher protein intake may also benefit endurance athletes because of their increased utilization of amino acids for fuel during long-duration activity and for recovery from muscle damage, especially during activities that may emphasize eccentric muscle actions, such as downhill running (i.e., especially for cross-country runners or marathoners [events included in the championships]) [15,17]. Intake of protein in the week before competition in amateur masters marathon runners was below guidelines, and a lower intake predicted worse performance [10]. Previous evaluations for masters athletes in other sports indicate a protein intake of 1.0–2.0 g/kg/d for males and 1.2–1.3 g/kg/d for females [4], similar to the recorded intakes in our study. The only other studies of elite masters track and field athletes, at the European or world championships, indicated low intakes of high-protein foods such as fish and dairy products [5,6].

Fat and saturated fat intakes of our athletes met the guidelines for athletes, which recommend <10% of calories from saturated fats and not less than 20% of calories from total fats, as adequate fat intake is important for the absorption of fat-soluble vitamins and the maintenance of cellular membranes [1]. Omega-3 fatty acid intake was relatively low in our athletes and this is in agreement with other studies of elite masters track and field athletes which determined low intakes of foods high in omega-3 (i.e., fatty fish) [5,6]. An intake of about 3 g/day of omega-3 and an omega-6-to-omega-3 ratio of 4 or less is recommended for masters athletes for the prevention of inflammation and the stimulation of muscle protein synthesis [2,3]. Given that fat intake was generally within guidelines, whereas carbohydrate and protein intakes were below guidelines for many of the athletes, it could be recommended to redistribute the macronutrient elements by reducing fat whilst increasing carbohydrate and protein intakes, or maintaining fat whilst increase carbohydrate and protein intakes.

The athletes in our study consumed a number of important micronutrients below the recommended levels. Calcium and vitamin D intakes, important for bone health, were well below guidelines for all athletic groups [2], and vitamin K intake, also important for bone health and blood clotting [18], was below guidelines for males [7]. With particular importance for endurance athletes, intakes greater than those in nutrition guidelines of about 2000 mg/d calcium and 1500–2000 IU/d vitamin D are effective for protecting against fatigue fractures [19]. Adequate calcium intake is important for all athletes for optimal muscle contraction and nerve conduction [1], and vitamin D may be important for stimulating muscle protein synthesis [1]. Vitamin K is important for proteins that inhibit osteoclast activation (i.e., cells involved in bone resorption) and promote bone mineralization [18]. Previous studies of masters track and field athletes at European or world championship athletic competitions indicate low intakes of calcium and vitamin D-rich foods such as fatty fish (i.e., for vitamin D) and dairy (i.e., for calcium and vitamin D) [5,6]. Previous evaluations of masters track and field athletes at world championships also indicated low intakes of vitamin K-rich foods including leafy green vegetables and broccoli [5]. A meta-analysis of dietary intakes of other masters athletes shows higher typical intakes of calcium (~1000 mg/d) [4], indicating that again our athletes may have found it challenging to find foods high in calcium and vitamin D while competing abroad.

Many of the athletes in our study had low intakes of antioxidant micronutrients. These included vitamins A, E, and C (in female endurance athletes), and zinc (in male and female endurance athletes). This is in agreement with a previous assessment of masters athletes at the European Athletic Championships which indicated low intakes of antioxidant-containing foods such as fruits and vegetables [5]. Sport nutrition guidelines recommend adequate intake of antioxidant micronutrients for the preservation of immune function [2] and the prevention of oxidative damage of cell membranes, especially in endurance athletes whose training elicits high rates of oxygen consumption [1]. Zinc is also important for energy metabolism, muscle contraction, and the regulation of ion channels at the neuromuscular junction [20], and low intake is associated with reduced aerobic power [21]. Foods high in zinc include peas, beans, lentils, chickpeas, nuts, seeds, chicken, and beef [22].

Additional micronutrients that were consumed below the recommended levels in our athletes included folate (except in female power athletes) and potassium. Folate is important, especially for endurance athletes, because it facilitates red blood cell production from bone marrow [23] and is therefore important with regards to hemoglobin and oxygen transport to muscles. Folate is found in leafy green vegetables, beans, peanuts, liver, fish, and eggs [23]. Previous assessments of masters athletes involved in world or European athletic championships indicated a low consumption of vegetables and fish [5,6]. Potassium is important for muscle and cardiac contractile performance, as it is an ion exchanged across muscle or cardiac membranes during repolarization [24]. Low plasma levels of potassium are predictive of muscle and whole-body fatigue [24]. Potassium is important for cardiovascular health, as it lowers blood pressure and is associated with reduced cardiovascular events [25]; therefore, adequate potassium intake is important for the health of older athletes. Potassium is high in fruits, vegetables, salmon, and dairy products, foods that were low in consumption from previous assessments of elite masters athletics competitors [5,6]. 

Although mean intakes met the recommended levels (Table 1), the proportion of endurance athletes consuming the recommended levels of vitamins B1 and B6 and iron were relatively low (Table 1). Vitamin B1 is important for energy metabolism, vitamin B6 is important for immune function, and iron is important for adequate hemoglobin levels and therefore oxygen delivery to muscles [22]. Meat, fish, and whole grains are good sources of B vitamins, and red meat and legumes are good sources of iron [22]. 

Alcohol intake was relatively low in the athletes from our study (i.e., mean of ~8 mg/d; Table 1), equivalent to about half a standard drink per day. This is important because alcohol impairs fat oxidation and decreases glycogen storage and muscle protein synthesis, all of which are important for athletic performance [1]. On the other hand, caffeine consumption at the level observed in our athletes (~3 mg/kg/d, Table 1) was at a level that may induce improvements in performance because caffeine acts as a central nervous system stimulant and blocks adenosine receptors and may increase the mobilization of fats from adipose tissue [26,27].

One limitation of our study was the classification of multi-sport athletes as power athletes. Three multi-sport athletes were included in our study: a male decathlete, a female heptathlete, and a female pentathlete. These events include mainly events that involve muscular power (sprinting, throwing, and jumping), but also include one “endurance”-type event (i.e., 1500 m in the decathlon or 800 m in the heptathlon and pentathlon). The nutritional needs of these athletes may be unique compared to athletes involved in events that strictly involve muscular power and this should be taken into consideration when interpreting the results of our study, especially regarding the female power group, because two of the five participants in this group were multi-sport athletes. 

A limitation of any analysis of dietary records is the underreporting of dietary intake by participants. It is estimated that underreporting with 24 h recalls is about 8–30%, which is similar to other methods for evaluating diet (i.e., underreporting for food records is about 11–41% and for food frequency questionnaires is about 5–42%) [28]. This may be the case in our study, as caloric intake assessed from the 24 h recalls was relatively low (Table 1). Low caloric intake in older adults may, however, be driven by reduced appetite caused by increased concentration of leptin (a hormone that reduces appetite) and decreased concentration of ghrelin (a hormone that increases appetite) [29,30]. Even if we assume that dietary intake was underreported by the upper amount for 24 h recalls (i.e., 30%), intakes of vitamin D, vitamin E, and vitamin A in males, zinc in male endurance athletes, and carbohydrates and calcium in endurance athletes would still be below the recommended levels for our athletes. There are no known studies that have evaluated the underreporting of dietary intake in masters athletes [4], and therefore, this is a direction for future research by comparing reported intakes to double-labeled water for energy expenditure. 

We were limited to using single 24 h recalls in our study because of participant burden during competition. Dietary records over multiple days (i.e., daily food logs) or multiple 24 h recalls may have provided greater accuracy for dietary intake. Single 24 h recalls are comparable to 3- or 4-day food records for mean intakes of energy and macronutrients (i.e., at the group level) [31,32], but not when assessed by Bland–Altman plots (i.e., at the individual level) [32]. The average of three 24 h recalls provides slightly better correlations, compared to single 24 h recalls, with the actual measures of energy intake by double-labeled water and protein intake assessed by daily urinary nitrogen [33]. The average of three 24 h recalls had correlations of 0.31 and 0.49 versus single 24 h recall correlations of 0.26 and 0.40 for actual energy and protein intake, respectively [33].

The dietary report in this study does not represent the athletes’ typical dietary routine, as the data were collected during competition. It is possible that most athletes may exhibit better nutritional habits outside of competitions due to better access to food. Future research should also consider what typical dietary habits masters athletes demonstrate, as this would then impact adaptation, ultimately dictating chronic performance. The dietary information collected during competition is likely indicative of acute performance changes. Directions for future research could include evaluating the effect of educational programs for masters athletes or the effect of increased high-nutrition food choices available onsite at championship events on the dietary intakes of the athletes.

## 5. Conclusions

Athletes at the World Masters Athletics Championships tend to consume amounts of protein and carbohydrates below the recommended levels to support optimal performance. The consumption of some important micronutrients, including calcium, vitamin D, and vitamin K, important for muscle and bone health were low. Many of the athletes had lower than recommended consumption of antioxidant micronutrients including vitamins A, E, and C, and zinc, important for immune function and protection against oxidative damage induced by high rates of oxygen consumption. Athletes had low intakes of folate and potassium, important for red blood cell production (and therefore oxygen delivery to muscles) and muscle and cardiac contractile performance, respectively. A low proportion of endurance athletes were consuming the recommended levels of vitamins B1 and B6 and iron, important for energy production, immune function, and hemoglobin production, respectively. Masters athletes would benefit from increased consumption of high-quality proteins (i.e., fish, lean meats, and dairy products), carbohydrates, and foods containing important micronutrients (i.e., zinc, vitamins A, B1, B6, C, D, E, and K, folate, potassium, and iron) which includes a variety of fruits and vegetables, legumes, nuts and seeds, whole grains, dairy products, red meats, and fish. Masters athletes could benefit from increased nutritional support when competing in a foreign environment.

## Figures and Tables

**Figure 1 nutrients-16-00564-f001:**
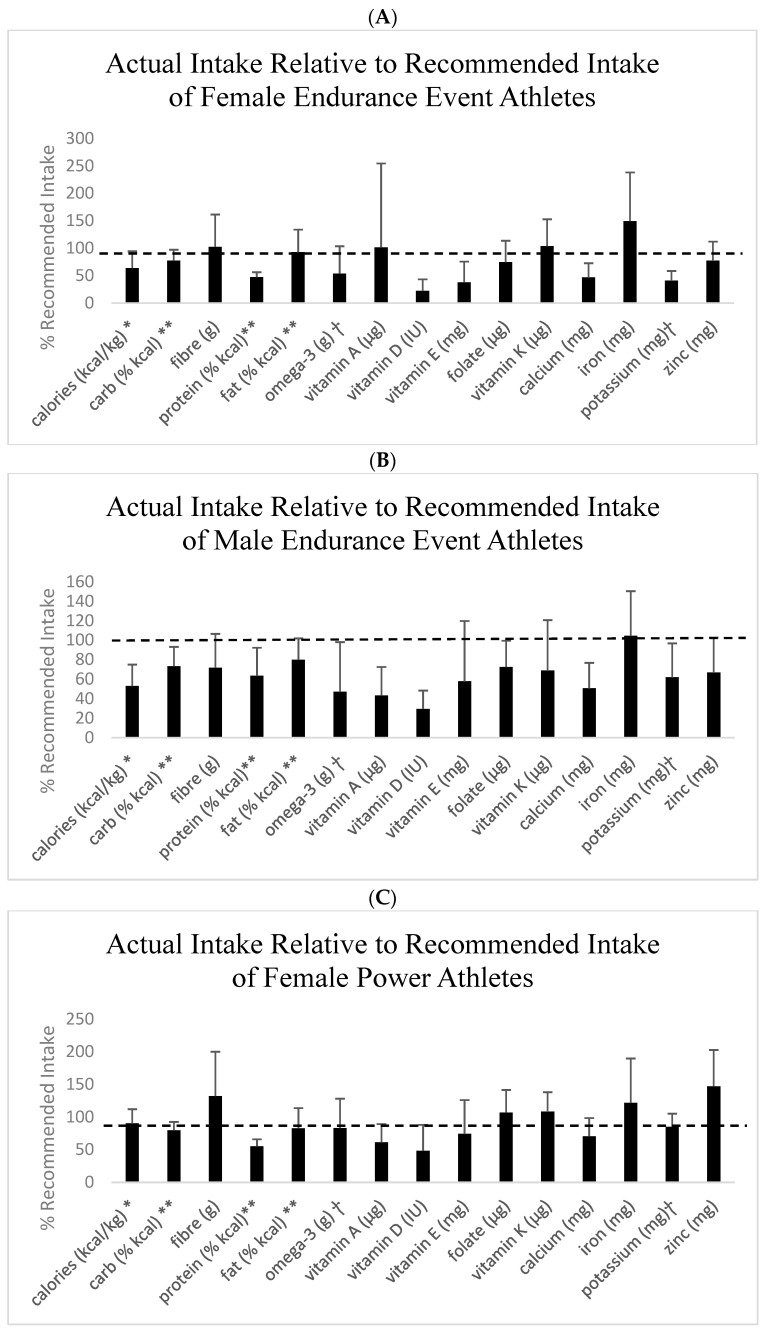
Dietary intakes relative to recommended intakes. Values are means and SD. Recommended intake is indicated as the dashed line at 100%. (**A**) Female endurance event athletes. (**B**) Male endurance athletes. (**C**) Female power athletes. (**D**) Male power athletes. * Population-specific recommendations [2]; ** acceptable macronutrient distribution range; ⴕ adequate intake. All other nutrients are expressed as a percentage of the recommended dietary intake.

**Table 1 nutrients-16-00564-t001:** Daily dietary intake of masters athletes by sex and demands of event.

Nutrient	Female Endurance (n = 11)	Male Endurance (n = 10)	Female Power (n = 5)	Male Power (n = 17)	Combined Athletes (n = 43)	Recommended Intake
Energy (kcal)	1585 ± 784	1546 ± 733	2293 ± 571 *	1989 ± 966 *	1818 ± 848	
Energy (kcal/kg)	29 ± 14 (45)	24 ± 10 (20)	41 ± 10 (80)	27 ± 12 (41)	28 ± 13 (42)	30–45 ^b^
Carbohydrates (g)	196 ± 106	188 ± 105	288 ± 39	232 ± 106	219 ± 102	
Carbohydrates (g/kg)	3.5 ± 1.8 (27) †	2.9 ± 1.5 (20)	5.1 ± 1.0 (60) †	3.1 ± 1.4 (6)	3.4 ± 1.6 (21)	5–10 ^a^
Total fiber (g)	22 ± 12 (45)	22 ± 10 (20)	30 ± 13 (80)	22 ± 11 (24)	23 ± 11 (35)	Females 21 Males 30
Protein (g)	64 ± 29	79 ± 50	107 ± 18 *	94 ± 44 *	84 ± 42	
Protein (g/kg)	1.1 ± 0.5 (27)	1.2 ± 0.8 (20)	1.9 ± 0.1 (100)	1.3 ± 0.5 (41)	1.3 ± 0.6 (40)	1.5 ^b^
Fat (g)	59 ± 37	51 ± 37	78 ± 50	70 ± 42	63 ± 40	
Fat (% kcal)	33 ± 14 (55)	28 ± 8 (90)	29 ± 11 (80)	31 ± 7 (76)	30 ± 10 (74)	20–35
Saturated fat (g)	20 ± 14	13 ± 5	22 ± 11	22 ± 15	19 ± 13	
Saturated fat (% kcal)	10 ± 4 (36)	8 ± 2 (90)	8 ± 2 (100)	9 ± 3 (53)	9 ± 3 (63)	<10 ^a^
Omega-3 Fatty Acid (g)	1.9 ± 1.7 (27)	1.6 ± 1.8 (10)	2.9 ± 1.6 (60)	2.5 ± 2.6 (24)	2.2 ± 2.1 (26)	3 ^b^
Omega-6 Fatty Acid (g)	10.7 ± 8.4	12.9 ± 18.9	12.1 ± 9.0	12.5 ± 9.9	12.1 ± 11.8	
Omega-6/omega-3 ratio	8 ± 7 (27)	7 ± 2 (20)	5 ± 2 (40)	8 ± 6 (18)	7 ± 6 (23)	4 ^b^
Vitamin A (µg RAE)	710 ± 1069 (27)	390 ± 265 (10)	428 ± 194 (20)	581 ± 421 (6)	552 ± 611 (14)	Females 700 Males 900
Vitamin B1 (mg)	1.1 ± 0.7 (36)	1.3 ± 0.5 (50)	2.1 ± 0.5 (100) *	1.7 ± 1.0 (65)*	1.5 ± 0.8 (58)	Females 1.1 Males 1.2
Vitamin B2 (mg)	1.7 ± 0.9 (82)	1.6 ± 0.5 (70)	2.4 ± 1.0 (100)	2.2 ± 1.1 (82)	1.9 ± 1.0 (81)	Females 1.1 Males 1.3
Vitamin B3 (mg)	21 ± 14 (64)	25 ± 14 (70)	29 ± 8 (100)	27 ± 13 (71)	25 ± 13 (72)	Females 14 Males 16
Vitamin B6 (mg)	2.2 ± 2.6 (45)	2.1 ± 1.3 (50)	2.9 ± 0.7 (100)	2.3 ± 1.0 (71)	2.3 ± 1.6 (63)	Females 1.5 Males 1.7
Vitamin B12 (µg)	6.1 ± 8.1 (64)	3.2 ± 1.7 (70)	4.6 ± 2.2 (100)	4.7 ± 3.1 (59)	4.7 ± 4.6 (67)	2.4
Vitamin C (mg)	57 ± 31 (27)	160 ± 136 (60)	150 ± 88 (60)	133 ± 112 (47)	122 ± 107 (47)	Females 75 Males 90
Vitamin D (IU)	136 ± 123 (0)	187 ± 114 (0)	289 ± 240 (20) *	277 ± 206 (6) *	221 ± 179 (5)	To 70 y = 600 71 y+ = 800
Vitamin E (mg)	6 ± 3 (0)	9 ± 9 (10)	11 ± 8 (0)	8 ± 6 (18)	8 ± 7 (9)	15
Folate (µg)	298 ± 157 (27)	290 ± 107 (20)	427 ± 139 (60) *	380 ± 167 (53) *	344 ± 153 (40)	400
Vitamin K (µg)	93 ± 44 (27)	82 ± 63 (20)	97 ± 27 (60)	91 ± 78 (53)	90 ± 61 (40)	Females 90 Males 120
Calcium (mg)	552 ± 313 (0)	529 ± 276 (0)	777 ± 366 (20)	783 ± 462 (29)	664 ± 385 (14)	Females <51, 1000 Females 51+, 1200 Males <71, 1000 Males 71+, 1200
Iron (mg)	12 ± 7 (55)	8 ± 4 (50)	14 ± 3 (40)	15 ± 9 (76)	13 ± 7 (60)	Females <51, 18 Females 51+, 8 Males 8
Potassium (mg)	1904 ± 831 (0)	2900 ± 1631 (0)	4000 ± 928 (40) *	3250 ± 1704 (18) *	2912 ± 1542 (12)	4700
Sodium (mg)	1703 ± 1257 (64)	1739 ± 1000 (80)	1950 ± 788 (80)	2108 ± 1452 (65)	1900 ± 1220 (70)	<2300 ^c^
Zinc (mg)	6 ± 3 (36)	7 ± 4 (10)	12 ± 4 (80)	11 ± 6 (29)	9 ± 5 (33)	Females 8 Males 11
Alcohol (g)	5 ± 7	2 ± 4	7 ± 9	14 ± 22	8 ± 15	
Caffeine (mg)	213 ± 155	223 ± 233	192 ± 195	159 ± 139	192 ± 172	
Caffeine (mg/kg)	3.7 ± 2.7	3.4 ± 3.4	3.3 ± 3.4	1.9 ± 1.8	2.9 ± 2.7	

All values are mean ± SD; the percentage of athletes meeting the recommended dietary intakes is reported in brackets (%); RAE = retinol activity equivalent; ^a^ based on sport nutrition guidelines [1]; ^b^ based on guidelines for masters athletes [2,3]; ^c^ upper limit. All other values are recommended dietary allowances/adequate intakes [8]. * Intakes are greater in power athletes compared to endurance athletes (*p* < 0.05). † Intake is greater in females compared to males (*p* < 0.05).

## Data Availability

Data are available upon request.

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
