# Peer review of "Dietary Intake of Athletes at the World Masters Athletics Championships as Assessed by Single 24 h Recall"

_nutrients, 2024, doi:10.3390/nu16040564_

Round 1

Reviewer 1 Report

Comments and Suggestions for Authors

Leonhardt et al. attempt to provide information regarding nutritional status of Master's Athletes during the World Masters Athletics Championships 2022/2023). The research is observational in nature, with no real statistical comparisons or interventions. In addition, it's significance is limited due to study design and lack of control (observational). My primary concerns are:

1. The dietary intake (primary outcome variable) is a 1-day recall. These are not particularly accurate and they can NOT be expected to reflect total nutrient intake over the course of the games. We would expect fluctuations for all nutrients over time. Ideally the authors would have planned this in advanced and asked participants to keep a daily food log (current record and not recall) over multiple days. I have a hard time finding any usefulness or validity in the use of a 1-day recall to reflect nutritional status of these athletes.

2. The graphs should include SD error bars, and possibly even a reference/recommendation line for easier comparison of actual vs recommended nutrient intakes.

3. At least for the sake of clarity, the authors should include their use of a 1-day dietary recall in their title and throughout their discussion. The design severely limits your interpretation and you should be clear of that throughout the paper.

Comments on the Quality of English Language

A few run-on sentences and minor English editing.

Author Response

Leonhardt et al. attempt to provide information regarding nutritional status of Master's Athletes during the World Masters Athletics Championships 2022/2023). The research is observational in nature, with no real statistical comparisons or interventions. In addition, it's significance is limited due to study design and lack of control (observational). My primary concerns are:

Response: In response to another reviewer’s comments, we have added statistical analyses between athletic groups and sexes for dietary intakes (Table 1, lines 84-90, and lines 109-119)

  1. The dietary intake (primary outcome variable) is a 1-day recall. These are not particularly accurate and they can NOT be expected to reflect total nutrient intake over the course of the games. We would expect fluctuations for all nutrients over time. Ideally the authors would have planned this in advanced and asked participants to keep a daily food log (current record and not recall) over multiple days. I have a hard time finding any usefulness or validity in the use of a 1-day recall to reflect nutritional status of these athletes.

Response: We have added discussion on this in our limitations section. We were limited by the participant burden imposed on athletes while competing at the games. We added discussion on the limitation of using a single recall versus daily food records or multiple 24-hour recalls. In summary, a single 24-hour recall compares well with multiple-day food records for mean intake of nutrients (i.e., good agreement at the group level) but not at the individual level (i.e., as determined by Bland-Alman plots). When comparing single versus 3-day 24-hour recalls, a 3-day 24 hour recall provides slightly better correlations with energy intake determined by doubly-labelled water and protein intake determined from 24-hour urinary nitrogen compared to a single 24-hour recall. Please see lines 299-309).

  1. The graphs should include SD error bars, and possibly even a reference/recommendation line for easier comparison of actual vs recommended nutrient intakes.

Response: We have added SD error bars and a line showing 100% of recommended intake for reference, as suggested.

  1. At least for the sake of clarity, the authors should include their use of a 1-day dietary recall in their title and throughout their discussion. The design severely limits your interpretation and you should be clear of that throughout the paper.

Response: We have changed the title as suggested. We have added at the beginning of the discussion section that dietary intake was assessed by single 24-hour recall. We have added a section at the end of the discussion on limitations of using a single 24-hour recall.

Reviewer 2 Report

Comments and Suggestions for Authors

I suggest changing the figures so that the groups are shown similarly to the tables: Figure 1 shows intakes relative to 100% with respect to gender. I suggest to divide groups in female endurance, male endurance, female power, male power to have a comparison between disciplines and sexes.

The order in the figure should be the same as in the tables.

I don't understand the ** in the Figure. What does potassium ** mean? Acceptable macronutrient distribution, but potassium intake is too low (in the results, you state that potassium intake was too low for all groups)?  Please explain in text or legend.

line 135: please mention the consumption for men and women for comparison with the literature (4)

Author Response

Thank you for the review of our manuscript and your comments. We have responded to all comments below. Revisions to the manuscript are highlighted.

I suggest changing the figures so that the groups are shown similarly to the tables: Figure 1 shows intakes relative to 100% with respect to gender. I suggest to divide groups in female endurance, male endurance, female power, male power to have a comparison between disciplines and sexes.

Response: We have divided the groups into separate figures, as suggested.

The order in the figure should be the same as in the tables.

Response: We have re-arranged the nutrients in the figure so they follow the same order as the table

I don't understand the ** in the Figure. What does potassium ** mean? Acceptable macronutrient distribution, but potassium intake is too low (in the results, you state that potassium intake was too low for all groups)?  Please explain in text or legend.

Response: The ** beside potassium in the figure was incorrect. We have removed this.

line 135: please mention the consumption for men and women for comparison with the literature (4).

Response: We have mentioned the intakes for men and women separate, as suggested by the reviewer (lines 172-174 in the revised manuscript).

Reviewer 3 Report

Comments and Suggestions for Authors

General comments

This study examined the dietary intake of master’s athletes that competed at the World Masters Athletics Championships. The study overall reported that the protein intake of the masters’ athletes were lower than recommended levels for this population, except for females in power-based sports. Several other vitamins were identified to be of lower than recommended levels for all athletes assessed. Thus, athletes competing at the world championships do not appear to consume sufficient levels of carbohydrate, protein and micronutrients. This study is of importance to determine whether athletes are meeting their dietary needs when competing at an international level and is of value to nutritionists and dietitians who monitor the dietary intake of athletes. Several comments have been provided which may enhance the quality of this paper.

Introduction

Lines 35-38: Increased requirement for protein, antioxidant, vitamin D and calcium compared to whom? Do you mean greater than? Just need to contextualize what this means.

Lines 47-50: This sentence is hard to follow. In the first half of the sentence, you have indicated paucity in research regarding masters’ athletes in athletics competitions, which makes sense. The latter half of the sentence almost seems to contradict the first half of the sentence. Do you mean that some studies have reported low dietary intake in male athletes across a range of sports?

Materials and Methods

Line 66: What classification have you used to classify running events of greater than 800m as endurance? Is there a reference for this?

Lines 68-70: Is it appropriate to classify multi-event athletes as power athletes, as you correctly state, that they are also involved in aerobic events? I would suspect that the energy and nutritional needs would be distinct between these athletes and those involved in purely power-based events.

Results

Table 1: The descriptive data provides the breakdown of various nutritional elements, separated by each athlete type, and is easy to follow. It would simplify the reporting if you could amalgamate Tables 1 and 2, so that you report the proportion of athletes that have met the recommended guidelines in brackets, after the standard deviation.

Figure 2: It is not clear what is being reported in this figure. Can you please provide more description, as not sure if this is proportion of the number of athletes, or supposed to be other values?

Overall: Have you considered conducting basic statistics to determine significant differences between these athletes for each nutritional element, such as a Kruskall Wallis test for continuous parameters? Also, a chi-squared test for comparing proportions of athletes that have, and have not met recommended guidelines?

Discussion

Lines 111-113: I would be cautious of this sentence, as it reads as though all athletes did not meet recommended guidelines, which is not the case based on the proportions you have reported.

Lines 119-120: I am being a little picky here, but the term “especially” is being used twice in this sentence, which comes across repetitively.

Lines 118-137: This paragraph compares the findings to previous studies succinctly. It would be interesting to get the author’s speculation on the potential reason for such low levels of carbohydrate intake, as this may assist practitioners to monitor and gauge carbohydrate intake in athletes of all levels. A potential reason for lower fibre intake was provided in the following paragraph, it would help if this approach is given to low carbohydrate intake as well.

Lines 164-172: It is interesting that the energy intake for most athletes were close to recommended intake of 30kcal/kg. Given that carbohydrate and protein were below recommended levels, particularly carbohydrate, and that fat intake was clearly within recommended levels, or even the upper limit, would you assume that the athletes’ fat intake increased their overall energy intake. If so, would it appropriate to redistribute the macronutrient elements, by reducing fat, whilst increase carbohydrate and protein, or maintain fat whilst increase carbohydrate and protein. These would be great discussion points for practitioners.

Line 183-186: This sentence comes across a little convoluted. Is it possible to separate the sentences so that is clear as to what foods should be consumed to increase vitamin D and K? The recommended food for zinc was nicely summarized, please use this approach for vitamin D and K.

Lines 230-241: It may also help to report that the dietary report in this study also does not represent the athlete’s typical dietary routine, as the data was collected during the competition. It is possible that most athletes may exhibit better nutritional habits outside of competition environments due to better access. Future research should also consider what typical dietary habits masters athletes demonstrate, as this would then impact adaptation, ultimately dictating chronic performance. The dietary information collected during competition is likely indicative of acute performance changes.

Author Response

Thank you for the review of our manuscript and your comments. We have responded to all comments below. Revisions to the manuscript are highlighted.

General comments

This study examined the dietary intake of master’s athletes that competed at the World Masters Athletics Championships. The study overall reported that the protein intake of the masters’ athletes were lower than recommended levels for this population, except for females in power-based sports. Several other vitamins were identified to be of lower than recommended levels for all athletes assessed. Thus, athletes competing at the world championships do not appear to consume sufficient levels of carbohydrate, protein and micronutrients. This study is of importance to determine whether athletes are meeting their dietary needs when competing at an international level and is of value to nutritionists and dietitians who monitor the dietary intake of athletes. Several comments have been provided which may enhance the quality of this paper.

Introduction

Lines 35-38: Increased requirement for protein, antioxidant, vitamin D and calcium compared to whom? Do you mean greater than? Just need to contextualize what this means.

Response: We have re-worded this sentence to indicate “greater requirements compared to younger athletes” (lines 35-36)

Lines 47-50: This sentence is hard to follow. In the first half of the sentence, you have indicated paucity in research regarding masters’ athletes in athletics competitions, which makes sense. The latter half of the sentence almost seems to contradict the first half of the sentence. Do you mean that some studies have reported low dietary intake in male athletes across a range of sports?

Response: To clarify this, we divided this into two sentences: “Very few studies have evaluated dietary intake of masters athletes involved in athletics competitions. No studies have determined intakes of specific nutrients, but two have indicated relatively low intake of important foods, such as fruits, vegetables, whole grains, dairy, and fish in male athletes.” (lines 48-50)

Materials and Methods

Line 66: What classification have you used to classify running events of greater than 800m as endurance? Is there a reference for this?

Response: We have cited the following reference to answer this question:

Duffield, R., Dawson, B., & Goodman, C. (2005). Energy system contribution to 400-metre and 800-metre track running. Journal of sports sciences, 23(3), 299–307. https://doi.org/10.1080/02640410410001730043

We have added the following sentences: “The classification of events ≥ 800m as “endurance” is based on the finding that this is the race distance at which the aerobic energy system begins to predominate, with aerobic energy system contribution estimated at 60% for men and 70% for women [7]. On the other hand, the 400m was classified as a “sprint” as the anaerobic energy system predominates, with anaerobic energy system contribution estimated at 59% for men and 55% for women [7].” (lines 69-74)

Lines 68-70: Is it appropriate to classify multi-event athletes as power athletes, as you correctly state, that they are also involved in aerobic events? I would suspect that the energy and nutritional needs would be distinct between these athletes and those involved in purely power-based events.

Response: Our study only included three multi-event athletes (one decathlete male, one heptathlete female, and one pentathlete female). This may have however impacted the group classified as female power athletes, since there were only five total athletes in this group. We have recognized this as a limitation at the end of our discussion section. We have added the following statement to our limitations section:

“One limitation of our study was the classification of multi-sport athletes as power athletes. Three multi-sport athletes were included in our study: A male decathlete, a female heptathlete, and a female pentathlete. These events include mainly events that involve muscular power (sprinting, throwing, jumping), but also include one “endurance” type event (i.e., 1500m in the decathlon, 800m in the heptathlon and pentathlon). The nutritional needs of these athletes may be unique compared to athletes involved in events that strictly involve muscular power and this should be taken into consideration when interpreting the results of our study, especially the female power group because two of the five participants in this group were multi-sport athletes.” (lines 276-284)

Results

Table 1: The descriptive data provides the breakdown of various nutritional elements, separated by each athlete type, and is easy to follow. It would simplify the reporting if you could amalgamate Tables 1 and 2, so that you report the proportion of athletes that have met the recommended guidelines in brackets, after the standard deviation.

Response: The two tables have been amalgamated, as the reviewer suggested. The proportion of athletes that have met the recommended guidelines are now indicated in brackets, after the standard deviation.

Figure 2: It is not clear what is being reported in this figure. Can you please provide more description, as not sure if this is proportion of the number of athletes, or supposed to be other values?

Response: The figure presents the mean intake of each nutrient relative the recommended intake. For example, the recommended fibre intake for females is 21g/d. The female endurance athletes on average consumed 22g/d, so in the figure, their % of Recommended Intake is slightly over 100%. We have added greater description in the figure legend to clarify this.

Overall: Have you considered conducting basic statistics to determine significant differences between these athletes for each nutritional element, such as a Kruskall Wallis test for continuous parameters? Also, a chi-squared test for comparing proportions of athletes that have, and have not met recommended guidelines?

Response: We have run some basic statistics as suggested. We used a 2-factor ANOVA to determine differences between athletic groups and sexes (and any interactions). We ran chi-squared tests to make comparisons between groups for proportion of athletes who met recommended guidelines. This is now described in the statistics section and results section of the manuscript. (lines 84-90, and lines 109-119)

To summarize: Power athletes had greater intake of calories, protein, folate, potassium, vitamin B1, and vitamin D compared to endurance athletes. There was only one sex main effect: females had greater carbohydrate intake relative to kg compared to males.

For chi-squared tests, there was a difference in proportions of athletes from different groups who met the recommended guidelines for all nutrients, except vitamins B2 and sodium. With some exceptions, for most of the nutrients, female power athletes had the greatest proportion of athletes meeting the recommended intakes.

Discussion

Lines 111-113: I would be cautious of this sentence, as it reads as though all athletes did not meet recommended guidelines, which is not the case based on the proportions you have reported.

Response: We have added the words “on average” to indicate that on average, athletes did not meet the recommended guidelines for specific nutrients. (line 148)

Lines 119-120: I am being a little picky here, but the term “especially” is being used twice in this sentence, which comes across repetitively.

Response: We have divided this into two sentences and deleted the second occurrence of the word “especially”. (line 157)

Lines 118-137: This paragraph compares the findings to previous studies succinctly. It would be interesting to get the author’s speculation on the potential reason for such low levels of carbohydrate intake, as this may assist practitioners to monitor and gauge carbohydrate intake in athletes of all levels. A potential reason for lower fibre intake was provided in the following paragraph, it would help if this approach is given to low carbohydrate intake as well.

Response: We have added the following to the end of the paragraph: “We can only speculate on why carbohydrate intake during competition was relatively low. This may be due to limited high-carbohydrate food available at the stadium or perhaps that the athletes had insufficient time to consume high carbohydrate foods while competing.” (lines 176-179)

Lines 164-172: It is interesting that the energy intake for most athletes were close to recommended intake of 30kcal/kg. Given that carbohydrate and protein were below recommended levels, particularly carbohydrate, and that fat intake was clearly within recommended levels, or even the upper limit, would you assume that the athletes’ fat intake increased their overall energy intake. If so, would it appropriate to redistribute the macronutrient elements. These would be great discussion points for practitioners.

Response: We would argue that the energy intake (g/kg) for most athletes was actually below recommended levels, except for female power athletes. For example, from Table 1, each group (except female power athletes) had mean intakes below the recommended levels and the proportion of athletes meeting recommended levels ranged from 20-45%. The reviewer makes a good recommendation thought that intake of fat could be reduced or maintained and a shift made to increase protein and carbohydrate intake. Throughout the discussion we have recommended that athletes be provided with greater amounts of high-carbohydrate and high-protein foods. In this paragraph on fat intake, we have added that it might be ideal to redistribute the macronutrient elements, by reducing fat, whilst increase carbohydrate and protein, or maintain fat whilst increase carbohydrate and protein. (lines 213-217).

Line 183-186: This sentence comes across a little convoluted. Is it possible to separate the sentences so that is clear as to what foods should be consumed to increase vitamin D and K? The recommended food for zinc was nicely summarized, please use this approach for vitamin D and K.

Response: We have separated this into two sentences for clarity. (lines 230-232)

Lines 230-241: It may also help to report that the dietary report in this study also does not represent the athlete’s typical dietary routine, as the data was collected during the competition. It is possible that most athletes may exhibit better nutritional habits outside of competition environments due to better access. Future research should also consider what typical dietary habits masters athletes demonstrate, as this would then impact adaptation, ultimately dictating chronic performance. The dietary information collected during competition is likely indicative of acute performance changes.

Response: We have added this near the end of our discussion section (lines 310-316)

Round 2

Reviewer 1 Report

Comments and Suggestions for Authors

My concerns were addressed

Reviewer 2 Report

Comments and Suggestions for Authors

Thank you for considering my remarks.